# Semi-supervised learning with Noisy Students improves domain generalization in optic disc and cup segmentation in uncropped fundus images

**Eugenia Moris**        EMORIS@PLADEMA.EXA.UNICEN.EDU.AR
**Ignacio Larrabide**        LARRABIDE@EXA.UNICEN.EDU.AR
**José Ignacio Orlando**        JIORLANDO@PLADEMA.EXA.UNICEN.EDU.AR
*UNCPBA, CONICET, Yatiris Group, Instituto Pladema, Tandil, Buenos Aires, Argentina.*

**Editors:** Accepted for publication at MIDL 2024

## Abstract

Automated optic disc (OD) and cup (OC) segmentation in fundus images has been widely explored for computer-aided diagnosis of glaucoma. However, existing models usually suffer from drops in performance when applied on images significantly different than those used for training. Several domain generalization strategies have been introduced to mitigate this issue, although they are trained and evaluated using images manually cropped around the optic nerve head. This operation eliminates most sources of domain variation, therefore overestimating their actual ability to cope with new, unseen patterns. In this paper, we analyze the most recent and accurate methods for domain generalization in OD/OC segmentation by applying them on uncropped fundus pictures, observing notorious degradations in their performance when trained and evaluated under this setting. To overcome their drawbacks, we also introduce a simple semi-supervised learning approach for domain generalization based on the Noisy Student framework. Using a Teacher model trained on a combination of domains, we pseudo-labeled a dataset of 18.000 originally unlabeled images that are then used for training a Student model. This semi-supervised setting allowed the Student network to capture additional sources of variability while retaining the original cues and patterns used by the Teacher through the weak annotations. Our results on eight different public datasets show improvements in every unseen domain over all alternative methods, and are available in https://github.com/eugeniaMoris/Noisy_student_ODOC_MIDL_2024.
**Keywords:** Domain Generalization, Semi-supervised learning, Segmentation

## 1. Introduction

Segmenting the optic disc (OD) and cup (OC) in fundus images is a common practice for detecting and characterizing glaucoma, one of the leading causes of irreversible blindness worldwide (Veena et al., 2020). A significant effort has been made to automate this task, resulting in models with excellent performance in known databases (Alawad et al., 2022; Moris et al., 2023; Wang et al., 2019). Nevertheless, their accuracy is frequently affected when applied on images from domains unseen during training, hampering their clinical application (Nan et al., 2022). This is inherent to the natural diversity in the appearance of fundus images, e.g. due to the overall quality of the scan, variations in the acquisition protocol or device, the intrinsic retinal pigmentation associated to patient ethnicity, or the presence of lesions that were not considered in the training sets (Yoon et al., 2023).

Typically, practitioners aim to overcome this limitation with increased data augmentation (Lyu et al., 2022), although modelling every possible scan (and disease) appearance

with image transformations is unfeasible. Alternatively, several studies introduced novel domain generalization techniques (Yoon et al., 2023) that aim to improve OD/OC segmentation in unseen domains without requiring domain-specific information or adaptation, e.g. through domain alignment (Chen et al., 2021; Liu et al., 2021; Wang et al., 2020; Chen et al., 2022; Hu et al., 2022; Zhou et al., 2022), meta-learning (Hu et al., 2023), and augmentation techniques (Lyu et al., 2022; Yang et al., 2021; Kang et al., 2022; Gu et al., 2023). While these have reported remarkable improvements when applied on unseen domains, we noticed that, in all cases, they are trained and validated using manual crops around the optic nerve head (ONH). This decision drastically eliminates most sources of image variability, implicitly overestimating their ability to cope with alterations outside this area. Furthermore, they also require users to perform the crop themselves when deployed in real clinical settings, hampering their automation and applicability for processing large databases.

In this paper we perform an in-depth evaluation of the most accurate existing models for domain generalization in OD/OC segmentation, by training and applying them on uncropped fundus images. In line with our hypothesis, we observe notorious degradations in their results when compared with their reported numbers. Furthermore, we show that a semi-supervised learning strategy based on the Noisy Student (Xie et al., 2020) is already able to overcome this limitation. Semi-supervised learning is an active area of research for medical image segmentation (Jiao et al., 2023), while weak supervision have shown promising results for OD segmentation e.g. through classification labels or bounding boxes in a multitask setting (Yin et al., 2023) and for source-free domain adaptation (Huai et al., 2023). In our case, we leverage a Teacher model trained on diverse domains to pseudo-label a dataset comprising 18.000 initially unlabeled images. This massive set is then used for training a Student model, which effectively assimilate new sources of variability while preserving the intrinsic cues and patterns imparted by the Teacher via weak annotations. Our evaluation across six unseen domains reveals consistent performance enhancements, surpassing the alternative methodologies. Furthermore, we also observe a statistical preservation of performance on unseen scans belonging to the three domains used for training.

## 2. Methods

### 2.1. Domain generalization with Noisy Students in uncropped images

Let $\mathcal{S} = \{(x_i, y_i)\}_{i=1}^n$ represent a dataset with $n$ pairs of uncropped fundus images $x_i \in \mathcal{X}$ and their corresponding multiclass OD/OC annotations $y_i \in \mathcal{Y}$. In a standard supervised learning setting, this set is used to train a multiclass segmentation model $f_{\theta_\mathcal{S}} : \mathcal{X} \to \mathcal{Y}$, with $f$ denoting the neural network architecture and $\theta_\mathcal{S}$ the learned parameters. When applying the model $f_{\theta_\mathcal{S}}$ on a new unseen target domain $\mathcal{D}_T$, it is desirable for $f_{\theta_\mathcal{S}}$ to retain its original performance. To this end, networks are usually trained in datasets as big and diverse as possible, e.g. by crafting $\mathcal{S}$ using subsets of images $\mathcal{X}$ sampled from multiple source domains $\mathcal{D}_S^{(j)}$. Notice that this requires scaling manual annotation for every new input sample, which is expensive and time consuming for segmentation tasks. As an alternative, one can stimulate the model to learn other alternative appearances through heavy data augmentation, although modelling every possible target scenario through image transformations is unfeasible. At the same time, it is likely that the model potentially

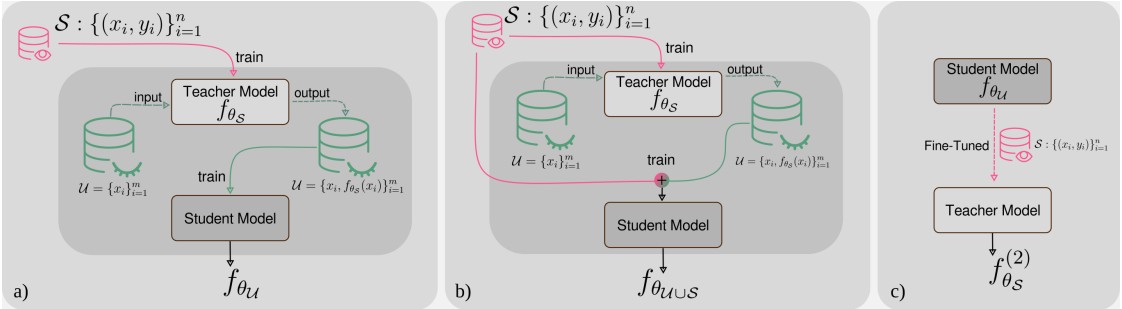

Figure 1: Schematic representation of our method. In a Noisy Student iteration, a Teacher $f_\theta^\mathcal{S}$ trained in a labeled set $\mathcal{S}$ is used to pseudo-annotate an unlabelled dataset $\mathcal{U}$, which is leveraged for training a Student, either (a) individually, or (b) jointly with $\mathcal{S}$. (c) The Student $f_{\theta_\mathcal{U}}$ could then by used as Teacher in a sequential manner by fine-tuning it on $\mathcal{S}$.

learns to cope artificially produced artifacts, as there are no guarantees about the actual resembling of these altered images to those expected from any unseen target domain $\mathcal{D}_T$.

Our hypothesis is that leveraging samples from a large enough unlabelled dataset $\mathcal{U}$ during training might mitigate and reduce the covariate shift between training and test domains, improving domain generalization with no extra manual annotation effort. Furthermore, feeding with uncropped fundus images should aid the network to capture sources of variation complementary to those around the ONH. To accomplish this, we propose to follow a Noisy Student approach (Xie et al., 2020), originally introduced for image classification. In this case, our goal is to learn a Student network for OD/OC segmentation in color fundus pictures using pseudo-labelled uncropped samples from $\mathcal{U}$. Figure 1 (a) depicts our methodology. First, we train a supervised Teacher model $f_{\theta_\mathcal{S}}$ using the labelled dataset $\mathcal{S}$, with uncropped images from a (combination of) source domain(s) $\mathcal{D}_S$. Afterwards, this network is applied on an unlabeled dataset $\mathcal{U} = \{x_i\}_{i=1}^m$, with $n << m$, to generate weak, pseudo-labels $\hat{y}_i$ of the OD and the OC. The resulting pairs $(x_i, \hat{y}_i)$ can then be used to train a Student model, either individually (resulting in $f_{\theta_\mathcal{U}}$, Figure 1 (a)), or in combination with $\mathcal{S}$ (resulting in $f_{\theta_{\mathcal{U} \cup \mathcal{S}}}$, Figure 1 (b)), (see Section 2.2). Both the Teacher and the Student are based on the exact same architecture $f$, and the Student is trained with stronger data augmentation than the Teacher, as suggested by Xie et al. (2020). This hardens the task solved by the Student model, forcing it to learn new patterns to obtain similar segmentations to those obtained by the Teacher, but in other different, more difficult images. This combination of new pseudo-labeled samples and data augmentation, and the fact that the Student is trained from scratch on this set, has been reported to improve results on out-of-distribution samples within seen test sets (Xie et al., 2020). Alternatively, we propose to apply this approach to improve results in unseen target domains $\mathcal{D}_T^{(j)}$.

## 2.2. Joint training vs. iterative fine-tuning and training

A Student $f_{\theta_\mathcal{U}}$ trained with pseudo-labels benefits from an implicit transfer of knowledge from the Teacher through weak targets. In practical terms, this should manifest in the Student having similar performance in the source domains $\mathcal{D}_S$. Nevertheless, recall that the Student is trained from scratch on $\mathcal{U}$, meaning that there was no direct access to images

from $\mathcal{S}$. Therefore, the model might behave in domains $\mathcal{D}_S$ as if they were target domains, experiencing a certain drop in performance. To alleviate this issue, one option is to include $\mathcal{S}$ in the training set of the Student, resulting in a model $f_{\theta_{\mathcal{U} \cup \mathcal{S}}}$ that had access to both sets simultaneously (Figure 1 (b)). Alternatively, we can think on an iterative process as the one in Figure 1 (c), in which the Student $f_{\theta_{\mathcal{U}}}$ is fine-tuned in $\mathcal{S}$ (resulting in $f_{\theta_{\mathcal{S}}^{(2)}}$) to become a Teacher, and then applied on $\mathcal{U}$ to produce new pseudo-labels to create a new $f_{\theta_{\mathcal{U}}^{(2)}}$ as in (Hao et al., 2022; Guan and Yuan, 2023). This process can be repeated $k$ times until reaching convergence, e.g. by evaluating performance increments on a held-out set.

## 3. Experimental setup

### 3.1. Implementation details

Code was implemented using PyTorch Lightning (v. 1.5.10), and all experiments were conducted using NVIDIA RTX 3060 GPUs with 12GB. Both Teacher and Student used the U-Net described in (Moris et al., 2023) (with 31 million parameters) as backbone. Training was performed minimizing a multiclass cross-entropy loss with Adam optimization. Learning rates were experimentally adjusted to each run, based on validation performance. Batch sizes of 20 images were used in all cases. For $f_{\theta_{\mathcal{S} \cup \mathcal{U}}}$, each batch included 18 samples from $\mathcal{U}$ and 2 from $\mathcal{S}$, to ensure the model had access to both domains on each iteration. To differentiate errors on each domain, we used a convex sum of losses with a $\lambda$ coefficient.

Data augmentation was used following a custom adaptation of RandAugment (Cubuk et al., 2020), using vertical and horizontal flipping, Gaussian blur, rotation, rescaling, and color jittering as transformations, parameterized with a probability $p$ of applying a transformation and a strength factor $s$ to increase or reduce their limits (e.g. angles in rotations, size of resizing and blurring filters, etc.). To avoid overfitting the Teacher, we also applied data augmentation for training it, using $p = 0.1$ and $s = 0.1$. The Student, on the other hand, was trained using $p = 0.5$ and $s = 0.5$. In all cases, images were resized to $256 \times 256$ pixels before feeding the network. In test time, output segmentation were re-scaled to the original image resolution using nearest neighbor interpolation for metric computation.

### 3.2. Materials and evaluation metrics

A summary of the datasets used for training, validation and test is provided in the appendix. We built the supervised training set $\mathcal{S}$ with images from DRISHTI (Sivaswamy et al., 2014), REFUGE (Orlando et al., 2020) and RIGA (Almazroa et al., 2018). In particular, we took all REFUGE training set (400 images), and 90% of DRISHTI training set (45 images) and RIGA (675 images). The remaining 10% of RIGA (74) and DRISHTI (5) were combined with the offsite set of REFUGE (400 images) to build a validation set. For $\mathcal{U}$, we randomly sampled 18000 scans from AIROGS (De Vente et al., 2023) training set. To calibrate model hyperparameters and monitor performance during training, 10% of these images were separated as a validation set. Test partitions from DRISHTI (50 images) and REFUGE (400 images) were used for testing in known domains. As unknown domains covering variations in acquisition machine, ethnicity, lesions, and field of view (FOV), we used all scans from RIM-ONE V3 (Fumero et al., 2011) (159 images, Spanish sample, cropped around the ONH), ORIGA (Zhang et al., 2010) (650 images from Malay adults), and the three subsets

Table 1: OC (top) and OD (bottom) segmentation results in uncropped images from unseen domains. The two best models are highlighted in bolds and underlined italics, respectively. Statistically significant improvements of Students $f_{\theta_\mathcal{U}}$ and $f_{\theta_{\mathcal{U} \cup \mathcal{S}}}$ are indicated with * and +, respectively.

| OC | | Method | RIMONE | BOSCH | FORUS | REMIDIO | ORIGA |
|---|---|---|---|---|---|---|---|
| ← DSC (%) | | Wang et al. (2020) | $64.32 \pm 30.71$ | $62.07 \pm 25.06^{*+}$ | $82.94 \pm 17.41^{*}$ | $61.07 \pm 25.49^{*+}$ | $71.90 \pm 22.52^{*+}$ |
| | | Chen et al. (2022) | $65.89 \pm 29.48$ | $66.22 \pm 20.74^{*+}$ | $84.95 \pm 9.99^{*}$ | $63.20 \pm 23.83^{*+}$ | $72.74 \pm 21.55^{*+}$ |
| | | Zhou et al. (2022) | $64.89 \pm 20.99^{*+}$ | $82.05 \pm 4.08^{*+}$ | $85.90 \pm 3.72^{*}$ | $80.58 \pm 6.14^{*+}$ | $76.48 \pm 12.02^{*+}$ |
| | | Teacher $f_{\theta_\mathcal{S}}$ | $54.38 \pm 22.87^{*+}$ | $\mathbf{87.71 \pm 5.08}$ | $83.64 \pm 8.54^{*}$ | $78.54 \pm 18.43^{*+}$ | $82.18 \pm 14.52^{+}$ |
| | | Ours $(f_{\theta_\mathcal{U}})$ | $68.39 \pm 19.32^{+}$ | $85.49 \pm 5.44$ | $\mathbf{88.85 \pm 5.17^{*}}$ | $\mathbf{86.25 \pm 6.92}$ | $81.08 \pm 11.07^{+}$ |
| | | Ours $(f_{\theta_{\mathcal{U} \cup \mathcal{S}}})$ | $\mathbf{70.37 \pm 18.23}$ | $85.52 \pm 5.94$ | $84.34 \pm 7.31^{*}$ | $85.83 \pm 8.87$ | $\mathbf{83.08 \pm 14.52}$ |
| HD → | | Method | RIMONE | BOSCH | FORUS | REMIDIO | ORIGA |
| | | Wang et al. (2020) | $44.78 \pm 28.72$ | $32.52 \pm 24.27^{*+}$ | $26.12 \pm 16.07$ | $64.92 \pm 27.50^{*+}$ | $46.34 \pm 23.93^{*+}$ |
| | | Chen et al. (2022) | $\mathbf{42.67 \pm 30.58}$ | $23.99 \pm 7.92^{*+}$ | $24.46 \pm 10.54$ | $58.82 \pm 22.28^{*+}$ | $47.41 \pm 24.33^{*+}$ |
| | | Zhou et al. (2022) | $64.11 \pm 34.26^{*+}$ | $15.67 \pm 5.65$ | $22.78 \pm 8.03$ | $38.95 \pm 14.55^{*+}$ | $64.75 \pm 154.56^{*+}$ |
| | | Teacher $f_{\theta_\mathcal{S}}$ | $77.97 \pm 40.12^{*+}$ | $14.83 \pm 4.67$ | $31.21 \pm 12.21^{*}$ | $90.10 \pm 140.43^{*+}$ | $49.28 \pm 110.03^{+}$ |
| | | Ours $(f_{\theta_\mathcal{U}})$ | $47.14 \pm 24.95$ | $\mathbf{14.78 \pm 4.48}$ | $\mathbf{22.14 \pm 7.47}$ | $\mathbf{34.00 \pm 14.42}$ | $\mathbf{36.08 \pm 18.24}$ |
| | | Ours $(f_{\theta_{\mathcal{U} \cup \mathcal{S}}})$ | $45.49 \pm 24.84$ | $17.51 \pm 5.44^{*}$ | $29.22 \pm 11.15^{*}$ | $34.51 \pm 16.78$ | $50.66 \pm 126.01$ |

| OD | | Method | RIMONE | BOSCH | FORUS | REMIDIO | ORIGA | PALM |
|---|---|---|---|---|---|---|---|---|
| ← DSC (%) | | Wang et al. (2020) | $78.64 \pm 29.03$ | $94.15 \pm 2.30^{*+}$ | $92.76 \pm 3.49^{*+}$ | $89.12 \pm 16.49^{*+}$ | $90.19 \pm 13.98$ | $74.71 \pm 36.56$ |
| | | Chen et al. (2022) | $82.43 \pm 21.54$ | $95.78 \pm 1.57^{*}$ | $94.46 \pm 2.16^{*+}$ | $91.58 \pm 12.23^{*+}$ | $89.64 \pm 14.52$ | $\mathbf{78.77 \pm 31.77}$ |
| | | Zhou et al. (2022) | $86.65 \pm 10.14$ | $92.22 \pm 7.60^{*+}$ | $93.20 \pm 2.02^{*+}$ | $90.20 \pm 9.64^{*+}$ | $90.18 \pm 9.97$ | $73.89 \pm 31.09$ |
| | | Teacher $f_{\theta_\mathcal{S}}$ | $81.59 \pm 20.03^{+}$ | $95.93 \pm 1.54^{*}$ | $95.94 \pm 2.53^{*+}$ | $90.65 \pm 10.39^{*+}$ | $89.98 \pm 11.55$ | $68.62 \pm 34.69$ |
| | | Ours $(f_{\theta_\mathcal{U}})$ | $86.88 \pm 7.72^{+}$ | $\mathbf{96.37 \pm 1.15}$ | $\mathbf{97.36 \pm 0.67}$ | $\mathbf{96.74 \pm 1.61}$ | $91.17 \pm 4.56$ | $68.03 \pm 35.37$ |
| | | Ours $(f_{\theta_{\mathcal{U} \cup \mathcal{S}}})$ | $\mathbf{87.11 \pm 7.04}$ | $96.19 \pm 1.05^{*}$ | $96.81 \pm 1.40^{*}$ | $95.13 \pm 3.12^{*}$ | $\mathbf{91.27 \pm 4.16}$ | $60.96 \pm 40.47^{*}$ |
| HD → | | Method | RIMONE | BOSCH | FORUS | REMIDIO | ORIGA | PALM |
| | | Wang et al. (2020) | $44.69 \pm 37.83$ | $12.59 \pm 4.19^{*}$ | $23.00 \pm 10.97^{*+}$ | $40.58 \pm 52.77^{*+}$ | $38.82 \pm 68.25$ | $\mathbf{59.72 \pm 129.64}$ |
| | | Chen et al. (2022) | $45.12 \pm 34.56^{*}$ | $\mathbf{10.37 \pm 2.0}$ | $20.48 \pm 7.37^{*+}$ | $36.77 \pm 31.49^{*+}$ | $42.41 \pm 60.69$ | $64.28 \pm 121.41$ |
| | | Zhou et al. (2022) | $50.77 \pm 43.56^{*+}$ | $22.94 \pm 84.53$ | $14.44 \pm 4.10^{*+}$ | $50.62 \pm 198.88^{*}$ | $54.84 \pm 88.22^{*+}$ | $133.58 \pm 209.77$ |
| | | Teacher $f_{\theta_\mathcal{S}}$ | $57.43 \pm 78.14^{*+}$ | $11.43 \pm 3.43$ | $16.57 \pm 7.64^{*+}$ | $71.17 \pm 112.90^{*+}$ | $46.99 \pm 80.98^{+}$ | $160.44 \pm 240.26$ |
| | | Ours $(f_{\theta_\mathcal{U}})$ | $\mathbf{35.31 \pm 18.22}$ | $11.38 \pm 2.95$ | $\mathbf{11.96 \pm 3.74}$ | $\mathbf{21.40 \pm 10.55}$ | $37.63 \pm 20.64^{+}$ | $165.22 \pm 252.30$ |
| | | Ours $(f_{\theta_{\mathcal{U} \cup \mathcal{S}}})$ | $39.26 \pm 21.58$ | $12.12 \pm 2.61$ | $12.78 \pm 3.01^{*}$ | $29.49 \pm 20.59^{*}$ | $\mathbf{35.80 \pm 15.15}$ | $182.88 \pm 255.38$ |

from CHAKSU (Kumar et al., 2023) (namely BOSCH–41 images–, FORUS–31 images–, and REMIDIO–264 images–, all from an Indian cohort and taken with 3 different devices). We also used images from PALM (Fang et al., 2024) (400 scans from an Asian population) to evaluate performance for OD segmentation in pathological myopia.

We used the Dice Similarity Coefficient (DSC) and Hausdorff distance (HD) as evaluation metrics, to account both for the overlap with the ground truth labels and for boundary consistency, respectively (Maier-Hein et al., 2022). Statistical significance of the differences in metrics was assessed using one-tail Wilcoxon sign-rank tests with $\alpha = 0.05$.

## 4. Results

Quantitative results for OC/OD segmentation in unseen domains are reported in Tables 1. We include results obtained with other domain generalization techniques that publicly released usable implementations for comparison. To ensure a fair evaluation, we re-trained their models with uncropped images, using our supervised set $\mathcal{S}$. Notice that none of them follows a semi-supervised learning approach. To our knowledge, there are no studies on domain generalization for OD/OC segmentation following this approach. Our Student $f_{\theta_\mathcal{U}}$ reported statistically significant improvements for OC segmentation with respect to the Teacher in terms of HD, for all the unseen domains, except BOSCH and ORIGA. This also holds for DSC values obtained in RIMONE, FORUS and REMIDIO. A similar behavior is

observed for OD segmentation, where this Student reported improvements over the Teacher in all sets except PALM evaluated in terms of DSC, and slightly higher HD values in BOSCH, with no statistically significant differences. On the other hand, the Student $f_{\theta_{\mathcal{U} \cup \mathcal{S}}}$ was able to statistically improve Teacher's DSC values for OC segmentation for all datasets except BOSCH and FORUS. When evaluated in terms of average HD, this Student also achieved significantly better OC segmentations in RIMONE, REMIDIO, and ORIGA, but less accurate results in BOSCH and FORUS. For OD segmentation, the Student $f_{\theta_{\mathcal{U} \cup \mathcal{S}}}$ showed a different behavior, reporting better DSC and HD values than the Teacher in all datasets except PALM, where average DSC and HD are lower but statistically comparable.

When comparing the Students one another, we observe that $f_{\theta_{\mathcal{U} \cup \mathcal{S}}}$ performs statistically significantly better than $f_{\theta_{\mathcal{U}}}$ in RIMONE and ORIGA when evaluated using DSC for OC segmentation, but comparable in BOSCH and statistically worse in FORUS and REMIDIO. When using the HD, on the other hand, $f_{\theta_{\mathcal{U}}}$ performs statistically better in BOSCH, FORUS, and ORIGA, almost equivalently in REMIDIO, and slightly worse in RIMONE. Conversely, for OD segmentation we see $f_{\theta_{\mathcal{U}}}$ reporting statistically better DSC values in FORUS, REMIDIO and PALM, and almost equivalent results in all other sets. In terms of HD, $f_{\theta_{\mathcal{U}}}$ reported values slightly better than $f_{\theta_{\mathcal{U} \cup \mathcal{S}}}$ in all sets except for ORIGA, although the differences are only statistically significant in FORUS and REMIDIO.

Compared with other existing approaches, both Students reported better DSC values for OC/OD segmentation in all the datasets, except for OD results in PALM. In terms of HD, we observe improvements over the literature for OC segmentation in all datasets except for RIMONE, and for OD segmentation in RIMONE, FORUS, REMIDIO and ORIGA. In PALM, both Students reported statistically higher HD values than the three evaluated counterparts, while in BOSCH the differences are not significant.

Figure 2 provides qualitative examples of OC/OD segmentation results obtained on images from different unseen domains. Examples were chosen to illustrate the behavior under changes in the acquisition device, ethnicity, lesions, and diversities of FOVs. As expected, the Teacher showed poor results in the unseen domains due to the intrinsic problem of domain generalization. Conversely, results obtained with the method by Zhou et al. (2022) demonstrate shape consistency throughout datasets, except under changes in ethnicity or when extensive peripapilary atrophy is present. Notice also a completely misfit of the OD with respect to the OC detection in the last image. Alternatively, our Students present more accurate segmentation for most of the cases, except for segmenting OC/OD in images with peripapillary atrophy when incorporating $\mathcal{S}$.

Finally, we performed additional experiments focused on comparing differences in seen and unseen domains (Figure A.1), evaluating the effect of sequentially repeating our framework for another iteration ($k = 2$, Table D.2 and D.3), and evaluating glaucoma detection results when using vertical cup-to-disc ratio estimates obtained from segmentations retrieved with the Teacher and the Student models (see Appendix C).

## 5. Discussion and Conclusions

Domain generalization remains being a challenge in OD/OC segmentation due to the significant variations observed between fundus pictures. While several approaches have been introduced to improve results under multiple imaging settings, we observed in our literature

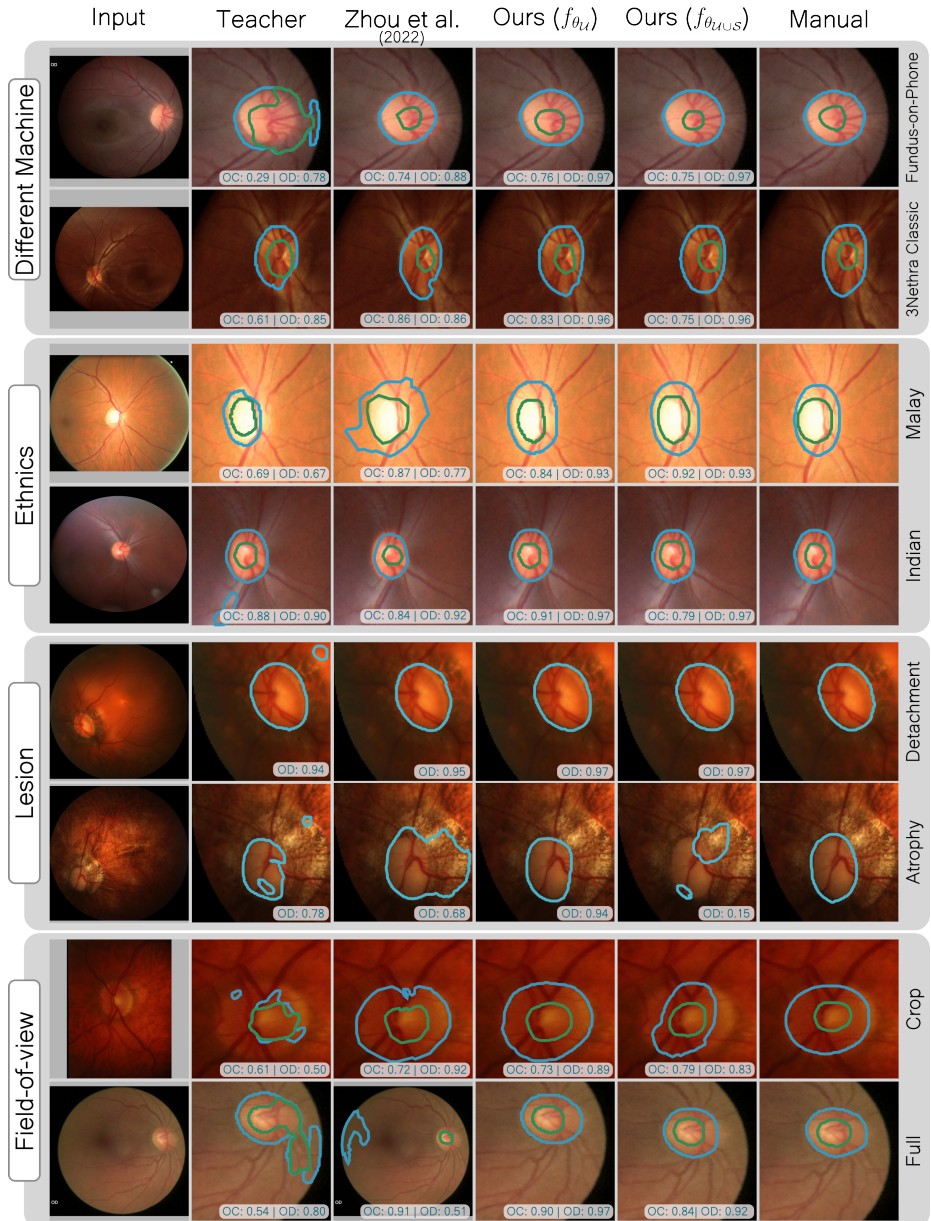

Figure 2: Qualitative results for OD (blue) and OC (green) segmentation in unseen domains, obtained by the Teacher, Zhou et al. (2022) method, and our proposed Students $f_{\theta_{\mathcal{U}}}$ and $f_{\theta_{\mathcal{U} \cup \mathcal{S}}}$ . Reference manual annotations and DSC values are included for comparison. Masks are zoomed for visualization purposes, although the input is in all cases an uncropped image.

review that all of them are trained and evaluated on cropped images around the ONH Wang et al. (2020); Chen et al. (2022); Zhou et al. (2022); Lyu et al. (2022); Yang et al. (2021). This drops out most sources of variability expected to occur in these images, reducing the space of patterns to model in training and overestimating their actual performance on full size images when testing. This can be observed in Table 1, where state of the art models

re-trained on uncropped scans obtained lower values than those reported in the original papers (Wang et al., 2020; Chen et al., 2022; Zhou et al., 2022). Additionally, training with cropped images introduces the need of doing a manual crop in test time, or training a separate model to automate it, e.g. in a coarse-to-fine manner (Moris et al., 2023). However, doing so was proved suboptimal for OD/OC segmentation if the coarse part lacks domain generalization capabilities (Moris et al., 2023). Thus, we are facing a causal loop paradox, where to achieve domain generalization we rely on a coarse detection model that needs to be able to generalize well to unseen images to ensure proper results.

In this paper we even show that a simple solution based on Noisy Students can outperform existing approaches. By training Students using a massive set of images pseudo-labelled by a pre-trained Teacher model, we obtained networks that reported better results in unseen domains than those obtained with other more technically complex methods trained and evaluated in full size images (Tables 1). These counterparts only reported better DSC and HD values for OD segmentation in one dataset, PALM, which features images of patients with pathological myopia. When analyzing results qualitatively, we observe that this drop is caused mostly when peripapilary atrophies are present (Figure 2), which significantly alters the appeareance of the boundaries of the OD. Another interesting remark observed in Figure A.1 is that following this approach does not degrade results in domains seen by the Teacher (or used also by the Student, if using $f_{\theta_{\mathcal{U} \cup \mathcal{S}}}$ ), statistically retaining most of the original performance. We also empirically showed that this behavior holds when using our segmentations for glaucoma detection using vCDR estimates (Figure C.2), reaching results in line or even superior to those obtained using manual segmentations.

Our results using Students with and without access to $\mathcal{S}$ were inconclusive, as both seems to be accurate in specific cases, sometimes reporting statistically comparable results. Considering this scenario, one could potentially combine their results e.g. in an ensemble setting, to take advantage of their individual results.

This Noisy Student framework is general enough to be applied with any backbone neural network architecture and/or domain generalization technique. In our experiments, we used a standard U-Net due to computational limitation, but other architectures with much more capacity could be leveraged to capture additional patterns Yi et al. (2023). Furthermore, our process could be leveraged in the context of any of the alternative approaches evaluated, potentially boosting their performance in unseen domains with uncropped images. Finally, notice that this technique could be extrapolated to any other fundus image segmentation task that requires domain generalization, e.g. diabetic retinopathy lesion segmentation. Nevertheless, as with any semi-supervised learning strategy based on weak labels, it must be considered that a poor Teacher model can degrade performance due to confirmation bias (Kwon and Kwak, 2022). Discarding this approach in advance would require to somehow approximate the performance in $\mathcal{U}$, which is challenging due to the intrinsic lack of labels on it. This particular limitation is an active research topic now, with many approaches being introduced e.g. to predict areas of error in the pseudo-labels or to rank labels based on the uncertainty of the model (Kwon and Kwak, 2022; Albert et al., 2023; Khan et al., 2024). Future work will focus on analyzing the potential contribution of these approaches in domain generalization results.

## Acknowledgments

This work was partially funded by Agencia I+D+i through PICT 2019-00070 and PICT startup 2021-00023, by CONICET through a PIP GI 2021-2023 - 11220200102472CO, and by an NVIDIA Hardware Grant.

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

Table .1: Summary with the number of samples used for training, validation and testing, indicating which datasets corresponded to seen and unseen domains.

| Dataset | Train | Validation | Test | Seen Domain | Unseen Domain |
|---|---|---|---|---|---|
| DRISHTI (Sivaswamy et al., 2014) | 45 | 5 | 50 | ✓ | ✗ |
| REFUGE (Orlando et al., 2020) | 400 | 400 | 400 | ✓ | ✗ |
| RIGA (Almazroa et al., 2018) | 675 | 74 | - | ✓ | ✗ |
| RIMONEv3 (Fumero et al., 2011) | - | - | 151 | ✗ | ✓ |
| ORIGA (Zhang et al., 2010) | - | - | 647 | ✗ | ✓ |
| BOSCH (Kumar et al., 2023) | - | - | 41 | ✗ | ✓ |
| FORUS (Kumar et al., 2023) | - | - | 31 | ✗ | ✓ |
| REMIDIO (Kumar et al., 2023) | - | - | 264 | ✗ | ✓ |
| AIROGS (De Vente et al., 2023) | 16200 | 1800 | - | ✓ | ✗ |

## Appendix A. Evaluation in seen domains vs. unseen domains

We performed an additional experiment comparing the performance in the seen domains obtained with the Teacher model and the Student approaches (see Table .1 for better readability). Figure A.1 depicts DSC and HD values for OD and OC segmentation in DRISHTI and REFUGE test sets, including also the unseen domains as a reference. Although differences are observed between the compared models, it is worth noting that the Students are statistically indistinguishable from the Teacher regardless the metric and the specific task.

## Appendix B. Evaluation of iteratively repeating the Noisy Student framework

We also performed an experiment evaluating the effect of sequentially repeating our framework for a second iteration ($k = 2$). Results are reported in Tables D.2 and D.3. For OC segmentation, we only observed improvements in the Teacher on FORUS, and REMIDIO. Retraining the Students on these labels did not improve results in any of the evaluated datasets, except in terms of HD in FORUS using $f_{\theta_{\mathcal{U}}}$ . For OD segmentation, on the other hand, the fine-tuned Teacher improves results with respect to the first version in all cases except for RIMONE. The re-trained Student reported statistically comparable results with respect to its one-iteration counterpart, with only slight improvements or decreases in their metrics. This analysis allows us to conclude that training for a second iteration is not worthwhile given that Students trained for just one iteration are more accurate.

## Appendix C. Evaluation of segmentation results for glaucoma assessment

We extended the evaluation with an experiment comparing glaucoma detection performance using manual segmentations and results obtained with the Teacher and our Noisy Student models. To this end, we computed ROC curves in both seen and unseen domains, using the vertical cup-to-disc ratio (vCDR) (Orlando et al., 2020) as a glaucoma score (Figure C.2). Notice that we did not include BOSCH and FORUS images in the evaluation as they only have one glaucomatous sample each. In line with our observations for overall segmentation accuracy, the Student models perform much better than the Teacher one in all unseen domains, with areas under the curve (AUC) that are comparable or even better than those obtained using ground truth segmentation. In seen domains like DRISHTI and

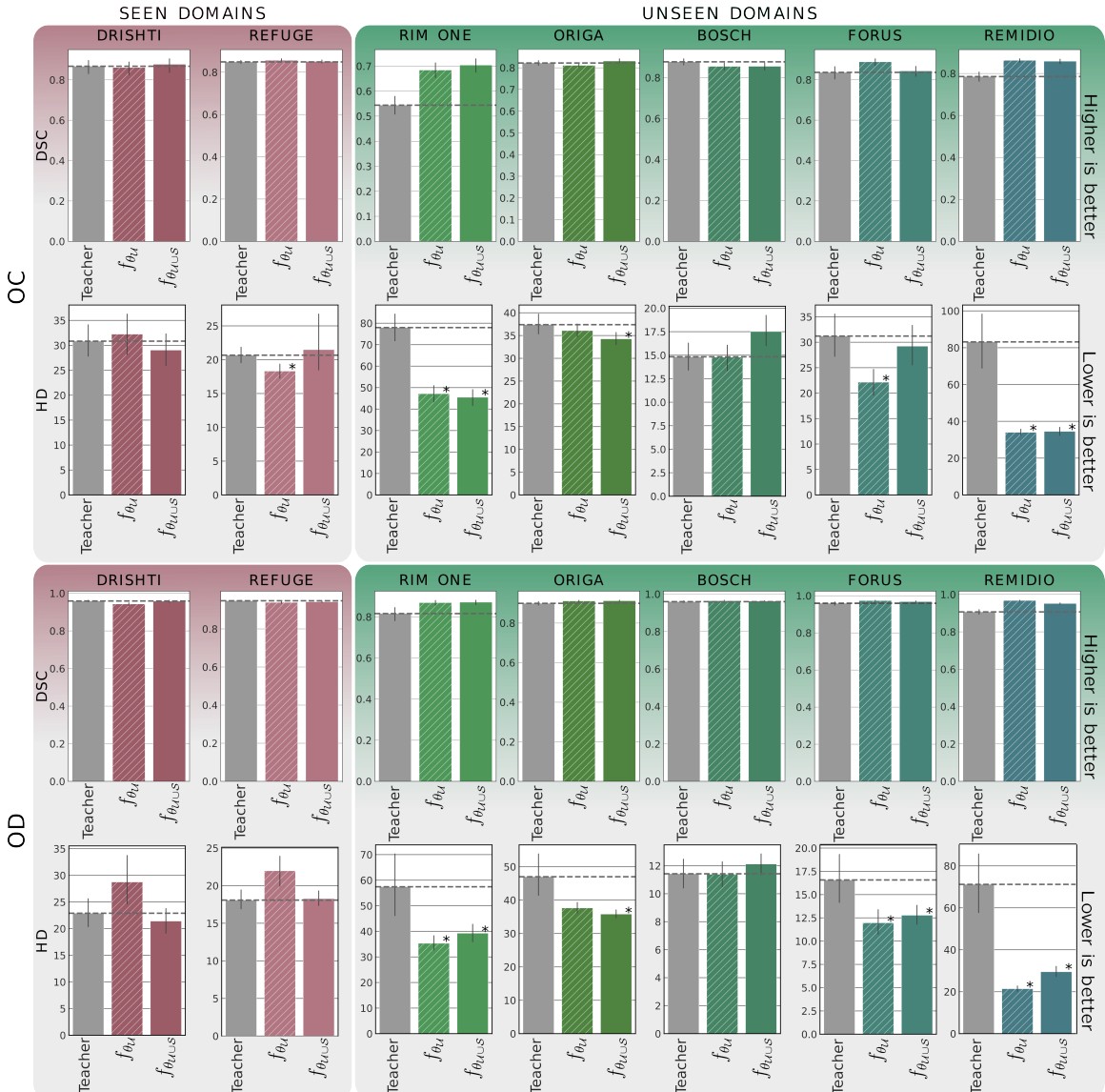

Figure A.1: DSC and HD values obtained for OC (top) and OD (bottom) in seen and unseen domains. The average value achieved by the Teacher is indicated as a dotted lines. * indicate statistically significant differences.

REFUGE, on the other hand, the Teacher model perform better than the Student without strong supervision ($f_{\theta_{\mathcal{U}}}$). Nevertheless, mixing both manual and pseudo-labelled scans (model $f_{\theta_{\mathcal{U} \cup \mathcal{S}}}$) reached classification results comparable to those obtained using manual segmentations.

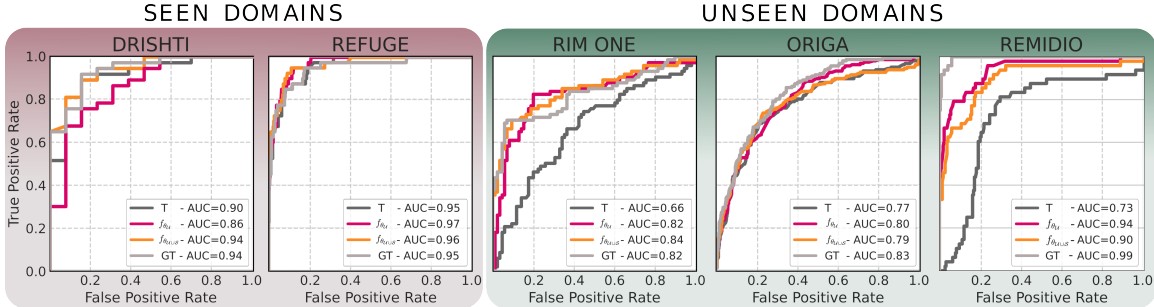

Figure C.2: ROC curves for glaucoma classification using vertical cup-to-disc ratios (vC-DRs) as glaucomatous scores, as derived from both ground truth (GT) segmentations and masks predicted with our Teacher (T) and Student ($f_{\theta_\mathcal{U}}$ and $f_{\theta_{\mathcal{U} \cup \mathcal{S}}}$ ) models, in both seen and unseen domains.

## Appendix D. Swapping the training set to CHAKSU

We evaluated also the performance of our Student models $f_{\theta_\mathcal{U}}$ and $f_{\theta_{\mathcal{U} \cup \mathcal{S}}}$ when using CHAKSU training sets as $\mathcal{S}$ for training a Teacher, and evaluating them all in all other sets as unseen domains. Results for OC and OD segmentation evaluated using DSC are depicted in Figure D.3. The Student model $f_{\theta_\mathcal{U}}$ improves results of the Teacher in almost all unseen datasets, except for OC segmentation in DRISHTI, although differences are not statistically significant. In seen domains, the model performs comparable to the Teacher, meaning that it is still able to retain its original performance. The Student $f_{\theta_{\mathcal{U} \cup \mathcal{S}}}$ , on the other hand, significantly improves results for OD segmentation in all datasets, both seen and unseen, and OC results in seen domains. In the unseen domains, however, OC segmentations in DRISHTI, REFUGE and ORIGA are worse than those obtained with the Teacher, experiencing a notorious drop in REFUGE and ORIGA.

Table D.2: OC segmentation results in uncropped images from unseen domains with $k = 2$ iterations. The two best models are highlighted in bolds and underlined italics, respectively. Statistically significant improvements of Students $f_{\theta_\mathcal{U}}$ and $f_{\theta_{\mathcal{U} \cup \mathcal{S}}}$ are indicated with * and +, respectively.

| OC | Method | RIMONE | BOSCH | FORUS | REMIDIO | ORIGA |
|---|---|---|---|---|---|---|
| DSC (%) ↓ | Teacher $f_{\theta_\mathcal{S}}$ | $54.38 \pm 22.87^{*+}$ | $\mathbf{87.71 \pm 5.08}$ | $83.64 \pm 8.54^{*}$ | $78.54 \pm 18.43^{*+}$ | $\mathit{82.18 \pm 14.52^{+}}$ |
| | **Ours** $(f_{\theta_\mathcal{U}})$ | $\mathit{68.39 \pm 19.32}$ | $85.49 \pm 5.44$ | $\mathbf{88.85 \pm 5.17}$ | $\mathbf{86.25 \pm 6.92}$ | $81.08 \pm 11.07$ |
| | **Ours** $(\mathcal{U} \cup \mathcal{S})$ | $\mathbf{70.37 \pm 18.23}$ | $\mathit{85.52 \pm 5.94}$ | $84.34 \pm 7.31$ | $\mathit{85.83 \pm 8.87}$ | $\mathbf{83.03 \pm 14.52}$ |
| | Teacher $(k = 2)$ | $53.22 \pm 26.27^{*+}$ | $79.05 \pm 9.28^{*+}$ | $\mathit{88.51 \pm 5.04}$ | $83.94 \pm 9.94^{*+}$ | $80.84 \pm 11.92^{+}$ |
| | **Ours** $(\mathcal{U}, k = 2)$ | $65.83 \pm 24.93$ | $74.23 \pm 7.81^{*+}$ | $86.42 \pm 8.43$ | $83.12 \pm 9.15^{*+}$ | $74.57 + 12.94^{*+}$ |
| HD → | Method | RIMONE | BOSCH | FORUS | REMIDIO | ORIGA |
| | Teacher $f_{\theta_\mathcal{S}}$ | $77.97 \pm 40.12^{*+}$ | $\mathit{14.83 \pm 4.67}$ | $31.21 \pm 12.21^{*}$ | $90.10 \pm 140.43^{*+}$ | $49.28 \pm 110.03^{+}$ |
| | **Ours** $(f_{\theta_\mathcal{U}})$ | $\mathit{47.14 \pm 24.95}$ | $\mathbf{14.78 \pm 4.48}$ | $\mathit{22.14 \pm 7.47}$ | $\mathbf{34.00 \pm 14.42}$ | $\mathbf{36.08 \pm 18.24}$ |
| | **Ours** $(f_{\theta_{\mathcal{U} \cup \mathcal{S}}})$ | $\mathbf{45.49 \pm 24.84}$ | $17.51 \pm 5.44$ | $29.22 \pm 11.15$ | $\mathit{34.51 \pm 16.78}$ | $50.66 \pm 126.01$ |
| | Teacher $(k = 2)$ | $98.41 \pm 181.39^{*+}$ | $18.07 \pm 4.78^{*}$ | $22.38 \pm 6.45$ | $35.94 \pm 15.53^{*+}$ | $\mathit{44.21 \pm 68.01^{*+}}$ |
| | **Ours** $(\mathcal{U}, k = 2)$ | $71.12 \pm 152.02$ | $20.27 \pm 4.13^{*+}$ | $\mathbf{21.47 \pm 7.51}$ | $38.65 \pm 19.86^{*+}$ | $47.11 \pm 42.47^{*+}$ |

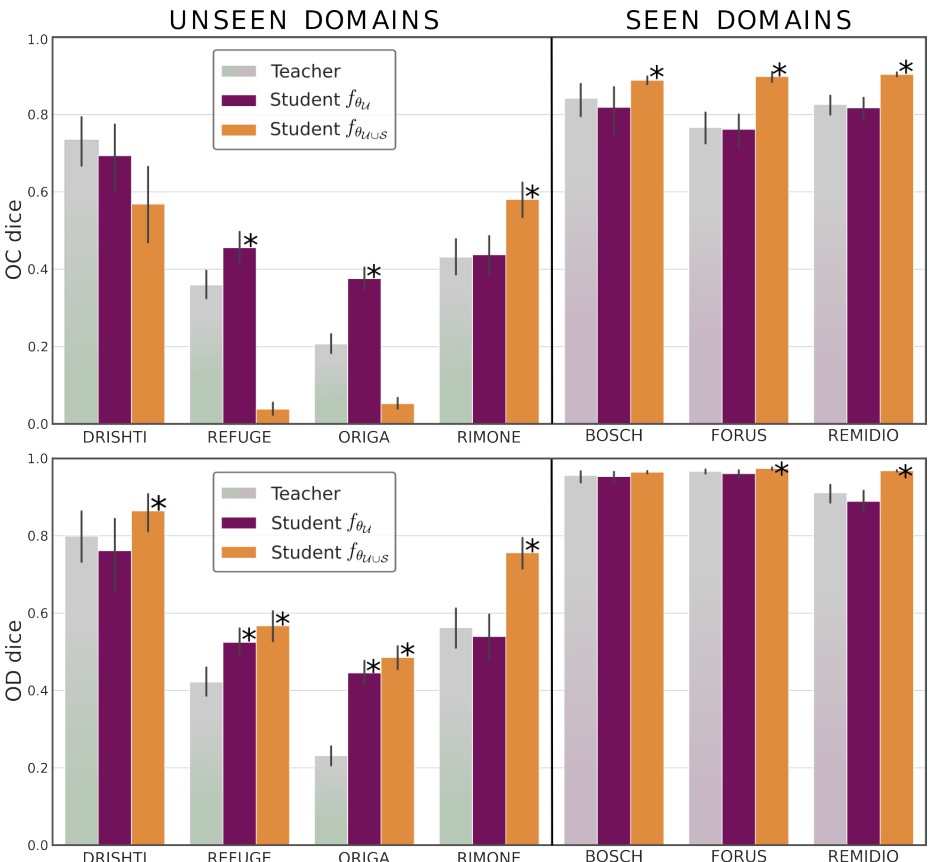

Figure D.3: Average dice (DSC) values obtained on unseen and seen domains when using CHAKSU training set for training our model. * indicate statistically significant differences.

Table D.3: OD segmentation results in uncropped images from unseen domains with $k = 2$ iterations. The two best models are highlighted in bolds and underlined italics, respectively. Statistically significant improvements of Students $f_{\theta_\mathcal{U}}$ and $f_{\theta_{\mathcal{U} \cup \mathcal{S}}}$ are indicated with * and +, respectively.

| OD | Method | RIMONE | BOSCH | FORUS | REMIDIO | ORIGA | PALM |
|---|---|---|---|---|---|---|---|
| DSC (%) ← | Teacher $f_{\theta_\mathcal{S}}$ | $81.59 \pm 20.03^+$ | $95.93 \pm 1.54^*$ | $95.94 \pm 2.53^{*+}$ | $90.65 \pm 10.39^{*+}$ | $89.98 \pm 11.55$ | *$68.62 \pm 34.69$* |
|  | **Ours** ($f_{\theta_\mathcal{U}}$) | *$86.88 \pm 7.72$* | $96.37 \pm 1.15$ | **$97.36 \pm 0.67$** | **$96.74 \pm 1.61$** | *$91.17 \pm 4.56$* | $68.03 \pm 35.37$ |
|  | **Ours** ($f_{\theta_{\mathcal{U} \cup \mathcal{S}}}$) | **$87.11 \pm 7.04$** | $96.19 \pm 1.05$ | *$96.81 \pm 1.40$* | *$95.13 \pm 3.12$* | **$91.27 \pm 4.16$** | $60.96 \pm 40.47$ |
|  | Teacher ($k = 2$) | $76.31 \pm 22.60^{*+}$ | *$96.66 \pm 0.97$* | $96.79 \pm 1.65^*$ | $95.75 \pm 3.79^*$ | $90.50 \pm 6.96$ | **$68.97 \pm 39.98^+$** |
|  | **Ours** ($\mathcal{U}$, $k = 2$) | $81.83 \pm 17.84$ | **$96.86 \pm 0.95$** | **$97.39 \pm 1.19$** | $96.49 \pm 3.24$ | $91.14 \pm 4.69$ | $60.96 \pm 40.47^+$ |
|  | Method | RIMONE | BOSCH | FORUS | REMIDIO | ORIGA | PALM |
| HD → | Teacher $f_{\theta_\mathcal{S}}$ | $57.43 \pm 78.14^{*+}$ | $11.43 \pm 3.43$ | $16.57 \pm 7.64^{*+}$ | $71.17 \pm 112.90^{*+}$ | $46.99 \pm 80.98^+$ | *$160.44 \pm 240.26$* |
|  | **Ours** ($f_{\theta_\mathcal{U}}$) | **$35.31 \pm 18.22$** | $11.38 \pm 2.95$ | **$11.96 \pm 3.74$** | **$21.40 \pm 10.55$** | $37.63 \pm 20.64$ | $165.22 \pm 252.30$ |
|  | **Ours** ($f_{\theta_{\mathcal{U} \cup \mathcal{S}}}$) | *$39.26 \pm 21.58$* | $12.12 \pm 2.61$ | $12.78 \pm 3.01$ | $29.49 \pm 20.59$ | **$35.80 \pm 15.15$** | $182.88 \pm 255.38$ |
|  | Teacher ($k = 2$) | $84.26 \pm 143.23^{*+}$ | *$10.26 \pm 2.34$* | $13.19 \pm 4.96^*$ | $24.89 \pm 35.32$ | $40.87 \pm 26.96$ | **$84.06 \pm 169.86^+$** |
|  | **Ours** ($\mathcal{U}$, $k = 2$) | $55.08 \pm 87.89^{*+}$ | **$9.76 \pm 2.20$** | *$12.04 \pm 7.58$* | *$23.35 \pm 21.04$* | *$36.72 \pm 19.31$* | $182.88 \pm 255.38$ |

