# OpenReview forum: "Semi-supervised learning with Noisy Students improves domain generalization in optic disc and cup segmentation in uncropped fundus images"
_MIDL.io/2024/Conference — MIDL 2024 Poster_

### Official Review · Reviewer_sVuo · 2024-02-28

**Confidence:** 5
**Preliminary Rating:** 3
**Recommendation:** Poster
**Final Rating:** 3.5

**Summary:**

In this study, the authors explored the concept of domain generalization for optic cup and disc segmentation in fundus images. They employed a semi-supervised technique with a noisy student to enhance the generalization ability. The authors utilized multiple datasets to train the algorithms and tested the model on studies that were not included in the training dataset, in order to evaluate the model's robustness.

**Strengths:**

The article's authors applied a. suggested technique to enhance the generalization of a trained model, which surpassed existing methods. They assessed the algorithm's robustness using standard metrics like the DICE similarityorff distance, which are reliable for algorithm evaluation. To train the algorithm and test its efficacy on various datasets, they utilized REFUGE, DRISTHTI, and RIGA. Although the semi-supervised with noisy student algorithm is commonly used in classification algorithms, the authors tested its effectiveness in segmenting cups and discs in fundus images. Overall, the article is well-written and easy to understand.

**Weaknesses:**

1. The authors chose REFUGE, DRISTHTI, and RIGA datasets for training and the rest for testing the algorithms. However, it is unclear how the algorithm performs if some of the datasets in training and testing sets are swapped.
2. There is limited novelty in the paper as the semi-supervised learning with noisy student is quite popular in the field of classification tasks and deep learning.

**Detailed Comments:**

The authors chose REFUGE, DRISTHTI, and RIGA datasets for training and the rest for algorithm testing. However, the algorithm's performance is unclear if some datasets in training and testing sets are swapped.  Including these improvements might be a very good paper.

**Justification Of Final Rating:**

The authors observed an improvement in performance on unseen datasets after swapping the datasets. The authors swapped the training set to CHAKSU, and the student model improved performance on all the unseen datasets. The paper includes new experiments that show the algorithm performs well on the new unseen domains.

**Justification Of The Preliminary Rating:**

Though the idea of using semi-supervised learning is a fairly popular idea, the authors have used this technique to improve domain generalization, which is one of the critical studies. I would suggest this paper for a poster.

**Questions To Address In The Rebuttal:**

NA

**Special Issue:**

No

---

> ### Author Response · Authors · 2024-03-16
>
> We thank the reviewer for taking their time to evaluate our manuscript. Below we address each of their comments:
>
> Weakness 1. The authors chose REFUGE, DRISTHTI, and RIGA datasets for training and the rest for testing the algorithms. However, it is unclear how the algorithm performs if some of the datasets in training and testing sets are swapped.
>
> We made this experimental choice to leverage the predefined training, validation and test partitions provided in REFUGE, DRISHTI and RIGA, using the non-partitioned ones for testing. Nevertheless, we perform an additional, complementary experiment swapping CHAKSU as a training set and using all the other test sets as unseen domains. These results are provided and discussed in Appendix D. Except for OC segmentation in DRISHTI, we observe that our Student models perform statistically better than the Teacher model in all unseen domains, while being comparable or even better than the Teacher in seen domains.
>
> Weakness 2. There is limited novelty in the paper as the semi-supervised learning with noisy student is quite popular in the field of classification tasks and deep learning.
>
> We agree with the reviewer in that leveraging Noisy Students is popular in the literature. Nevertheless, we believe ours is the first article applying it in the field of domain generalization for effective OD/OC segmentation from full size images. Furthermore, as discussed in the article, our goal was not to introduce a novel method but to show that, due to an experimental problem in the existing literature, most of technically novel approaches were not able to perform better than an already established semi-supervised learning baseline. We believe that the revised version of our manuscript better reflects this idea, e.g. thanks to our modifications in Sections 1, 2.1 and 5.

---

> > ### Comment · Reviewer_sVuo · 2024-03-20
> >
> > I thank the authors for their comprehensive response and for conducting additional experiments especially with swapping the datasets. Upon reviewing the results, I am convinced that the proposed method has significant advantages in domain generalization. I am pleased to raise my score accordingly.

---

### Official Review · Reviewer_g92B · 2024-02-28

**Confidence:** 3
**Preliminary Rating:** 4
**Recommendation:** Poster
**Final Rating:** 4

**Summary:**

This paper presents a “noisy student” framework for semi-supervised learning for domain generalization. A teacher model is first trained on a combination of all domains and used to pseudo label unlabeled images for training a student model. A key premise of the work is that many existing DG methods rely on manually cropping around the optic nerve head, which reduces most sources of domain variations and thus limiting their ability in real-world use scenarios. Experiments demonstrated clear improvements of the proposed approach on OC and OD segmentations in uncropped images across different datasets.

**Strengths:**

The proposed work highlights an interesting hidden gap in existing works, i.e., the reliance on manual cropping of images, that may have overestimated the ability of existing works in handling domain variability. This is insightful.

The proposed method is simple but demonstrated overall convincing performances over the six different datasets considered.

The paper was clearly written and easy to follow. The analyses of results were concrete and detailed pertaining to each dataset (rather than general statements of performance gains).

**Weaknesses:**

The baselines included in the comparison were not adequately discussed, especially whether they are semi-supervised nor not.

The proposed work is of limited technical innovation as it builds primarily on a previously published “noisy student” framework. This is not a major concern though as it’s offset by the insight provided and experimental results shown.

**Detailed Comments:**

It will be helpful to add more descriptions of the baselines included in the comparison, especially if there are differences in terms of whether they are all semi-supervised methods.

**Justification Of Final Rating:**

After going through my fellow reviewers' comments and the authors' response, I will maintain my earlier rating based on its interesting concept, promising empirical results, and the gap of research it is addressing in the current literature.

**Justification Of The Preliminary Rating:**

This is an overall well-written paper that identified a specific and interesting gap in the existing literature, and proposed a simple solution with compelling empirical evidence. Although the proposed method was mainly based on a previously published method, its ability to address the identified gap is convincing and worthy of publication.

**Questions To Address In The Rebuttal:**

See above.

**Special Issue:**

Yes

---

> ### Author Response · Authors · 2024-03-16
>
> We thank the reviewer for their feedback on our manuscript. We have modified our manuscript according to their comments and suggestions, namely:
>
> Weakness 1. The baselines included in the comparison were not adequately discussed, especially whether they are semi-supervised nor not.
>
> Unfortunately we were unable to include a proper description and analysis of each of the three considered approaches due to space constraints. Nevertheless, we modified the first paragraph in Section 4 to include this particular request from the reviewer, explaining that none of the compared methods follows a semi-supervised learning approach.
>
> Weakness 2. The proposed work is of limited technical innovation as it builds primarily on a previously published “noisy student” framework. This is not a major concern though as it’s offset by the insight provided and experimental results shown.
>
> We thank the reviewer for pointing out this fact. The goal of our manuscript is not to introduce a novel technical methodology but to demonstrate a significant experimental problem in the literature of domain generalization for OD/OC segmentation, and at the same time show that a simple approach such as the Noisy Student framework can surpass the existing literature if applied on an uncropped setting. Furthermore, ours is the first study showing that the Noisy Student approach not only improves results for out-of-distribution samples in seen domains as reported by Xie et al. 2020, but also improves in fully unseen domains.

---

### Official Review · Reviewer_XqLd · 2024-03-01

**Confidence:** 4
**Preliminary Rating:** 3
**Final Rating:** 3.5

**Summary:**

This work challenges the improvement of generalisation ability in the segmentation of optic disc and cup in fundus images. The authors indicate the existence of unfair manual preprocessing for these segmentation in the previous works' experimental evaluations, which prevents the evaluation of actual generalisation ability of a trained model. After omitting the unfair preprocessing in the segmentation problem, the authors apply the Noisy Student Framework to improve a model’s generalisatoin ability. In the experiments, the authors performed comparative evaluation with several open-access datasets.

**Strengths:**

-Indicating the problem of experimental evaluations of fundus image segmentation in previous works.

-Focusing the actual generalisation ability for fair evaluations of trained models in fundus image segmentation.

-Applying the Noisy Student Framework with its extension into the segmentation of optic disc and cup in fundus images.

-Using the several open-access datasets for experimental evaluations.

**Weaknesses:**

-The technical novelty of the proposed method is incremental

-The survey for weak-supervised learning with pseudo labels seems missing.

-Table summarising the details of training, validation and testing datasets is necessary for high readability.

**Detailed Comments:**

Even for medical-image segmentation problem, semi-supervised or weak supervised learning with pseudo annotation labels have already been proposed. To indicate the technical novelty of this work, the survey of these related works is necessary. Iterative approach for teacher and student models might appear in several previous works.

**Justification Of Final Rating:**

I thank the authors’ feedback. The authors gave feedback to my questions. Even though technical novelty is incremental, some of conference attendees might have interest on this work. Therefore, I rate this submission as borderline accept.

**Justification Of The Preliminary Rating:**

The concept and approach of this work are interesting. The efficient training strategy for the limited annotated data is essential problem in a machine learning application to medical images. Even though the technical novelty of this work seems incremental and the current survey of related works is unsatisfactory, the authors presented the fair evaluations even for the methods of previous works by using several datasets. Therefore, I conclude that this submission is in Borderline.

**Questions To Address In The Rebuttal:**

For practical application of OC/OD segmentations, how accurate segmentation models are required for each?

If the given pseudo annotations on U are accurate, i.e. the teacher model can accurately segment samples in U, this means that almost all samples in U have similar patterns to S. In this case, why a student model can achieve more accurate segmentation than a teacher model even though many patterns used in the training are almost the same in two sets?

If the given pseudo annotations on U are inaccurate, i.e. the teacher model can’t accurately segment sample in U, this means that many samples in U have different patterns from S. In this case, why larger noisy annotated training set can lead to a more-accurate segmentation model? Noises can lead to inaccurate segmentation model, where errors caused by noises are accumulated. Is there  theoretically convincing explanation?

---

> ### Author Response · Authors · 2024-03-16
>
> We thank the reviewer for the time and feedback on our article, and for pointing out its strengths. We analyzed all the comments and recommendations from the reviewer and modified our original manuscript to clarify and complete all missing points.
> Below we answer the specific concerns requested by the reviewer for this rebuttal:
>
> -R1. For practical application of OC/OD segmentations, how accurate segmentation models are required for each?
>
> OD/OC segmentations are clinically used for calculating glaucoma biomarkers such as the vertical cup-to-disc ratio (vCDR) or the disc damage likelihood scale (DDLS). Both features require accurate input segmentations for a proper estimate, yet there is no consensus about which Dice or Hausdorff Distance values ensure proper estimates of these features. Nevertheless, to illustrate the contribution of our domain generalization strategy to the final performance estimates, we included Appendix C illustrating the improvements in glaucoma assessment with vCDR values derived from segmentations obtained by applying our domain generalization technique (ROC curves in Figure C.2). We also updated Sec. 4 (par. 5) and 5 (par. 2) to refer to this incorporation.
>
> R2. If the given pseudo annotations on U are accurate, i.e. the teacher model can accurately segment samples in U, this means that almost all samples in U have similar patterns to S. In this case, why a student model can achieve more accurate segmentation than a teacher model even though many patterns used in the training are almost the same in two sets?
>
> Incorporating stronger data augmentation as noise when training the (Noisy) Student yields a model that differs from the Teacher. Data augmentation further hardens the task of the Student, forcing it to learn new, unseen patterns to keep producing results as those obtained with the Teacher model. If accurate labels are obtained in U using the Teacher, then the Student can increase this accuracy by being trained from scratch on this set of further transformed images. Xie et al. 2020 reported that this methodology improved results on out-of-distribution samples within seen domains. In our experiments, we show that this also holds for new samples from unseen domains, resulting in a valid alternative for reaching better domain generalization. We included a few lines in the last paragraph of Sec. 2.1 to summarize this rationale, and extended Sec. 5 with a new paragraph in the end, in line also with the comment R3.
>
> R3. If the given pseudo annotations on U are inaccurate, i.e. the teacher model can’t accurately segment sample in U, this means that many samples in U have different patterns from S. In this case, why larger noisy annotated training set can lead to a more-accurate segmentation model? Noises can lead to inaccurate segmentation model, where errors caused by noises are accumulated. Is there theoretically convincing explanation?
>
> Indeed, existing literature reports that having trustworthy annotations by the Teacher model in the set of unlabelled samples U used for semi-supervised learning is fundamental to ensure a good performance (e.g. Kwon and Kwak, 2022). However, as no annotations are available for U, it is not feasible for us to estimate the accuracy of these particular segmentations. To raise awareness about this particular topic, we extended Sec. 5 by incorporating a paragraph in the end referring to this observation, including also relevant literature and potential solutions.
>
> We also addressed comments and potential weaknesses indicated by the reviewer:
>
> W1. The technical novelty of the proposed method is incremental
>
> We agree that the novelty of our method is incremental, although we believe our article contributes to the state of the art from multiple perspectives. As indicated by the reviewer, we are reporting a significant experimental problem in the existing literature in domain generalization for OD/OC segmentation. Furthermore, we are extending the Noisy Student approach to this particular segmentation domain, illustrating that this already established method is able to outperform other more complex approaches from a domain generalization perspective. Finally, we are releasing our results publicly so that new methods in the literature can effectively compare their segmentations with ours in a reproducible manner.
>
> W2. The survey for weak-supervised learning with pseudo labels seems missing.
>
> We have extended Sec. 1 (par. 3) to the best of our ability considering the space limitation, including references to recent studies on weak supervision and semi supervised learning in the context of fundus image analysis applications.
>
> W3. Table summarising the details of training, validation and testing datasets is necessary for high readability.
>
> We included a table in the Appendix (Table .1) that summarizes the number of samples on each training, validation, and test partition for each dataset, indicating also which ones were considered as seen and unseen domains.

---

### Meta-Review · Area_Chair_yuKp · 2024-03-30

**Recommendation:** Accept (Poster)
**Confidence:** 5

**Metareview:**

This submission investigates the use of semi-supervised learning with noisy student models to enhance domain generalization in medical imaging. While the core technical novelty might be somewhat limited, the primary strength of the work lies in its extensive evaluation and insights into the effectiveness of this approach.

The reviews have a consensus on the limited novelty (builds on existing methods, limited novelty), but appreciate a strong evaluation (thorough evaluation across multiple datasets, demonstrating the value of the approach in a practical setting - a contribution for MIDL).

Despite the incremental technical novelty, the strong evaluation, the focus on domain generalization, and the interest expressed by reviews warrant an acceptance of this paper. It could present a valuable discussion for the MIDL community.

The authors are encouraged to strengthen the related work section to better contextualize their contribution in light of existing techniques.

For all these reasons, and with respect to the other submissions, the recommendation is towards Acceptance.

To PC: This is the best paper of my batch (my batch was possibly of a lower quality this year)

---

### Decision · Program_Chairs · 2024-04-05

Accept (Poster)